

# Knowledge, attitude and practice towards bio-medical waste management among healthcare workers: a northern Saudi study

Ashokkumar Thirunavukkarasu[1], Ahmad Homoud Al-Hazmi[1], Umar Farooq Dar[1], Ahmed Mohammed Alruwaili[2], Saleh Dhifallah Alsharari[2], Fahad Adel Alazmi[2], Saif Farhan Alruwaili[2] and Abdullah Mohammed Alarjan[2]

[1] Department of Community and Family Medicine, College of Medicine, Jouf University, Sakaka, Aljouf, Saudi Arabia
[2] Medical Student, College of Medicine, Jouf University, Sakaka, Aljouf, Saudi Arabia

## ABSTRACT

**Background.** Health care workers (HCWs) involved in day-to-day care and other healthcare activities play a significant role in biomedical waste (BMW) management. The World Health Organization stated some of the causes for the failure of BMW management, namely, lack of awareness of the health hazards related to BMW and inadequate trained HCWs in BMW management. The present study assessed knowledge, attitude, and practice (KAP) towards BMW management among the HCWs in northern Saudi Arabia.

**Methodology.** The present study assessed KAP through a self-administered and validated questionnaire. Using a multistage probability sampling method, 384 HCWs from different healthcare facilities participated in this survey. We performed binomial logistic regression analysis to find association between KAP subscales and sociodemographic characteristics. Spearman's correlation test was performed to find the strength and direction of correlation (*rho*) between KAP scores.

**Results.** Of the population studied, high knowledge, attitude, and practice scores were found in 47.1%, 65.1%, and 49.5% of the HCWs, respectively. The present study found that knowledge score was significantly higher among the age group from 30 to 39 years (ref: age less than 30 years: AOR = 2.25, 95% CI [1.05–4.85], $p = 0.04$) and non- Saudi nationals (ref: Saudi: AOR = 2.84, 95% CI [1.63–4.94], $p < 0.001$) The attitude score towards BMW management was significantly lower among the HCWs working in tertiary care settings (ref: PHC: AOR = 0.38, 95% CI [0.12–0.69], $p = 0.01$). Regarding the practice score, the male categories had a significantly higher score (ref: female: AOR = 1.82, 95% CI [1.19 –2.99], $p = 0.02$), while pharmacist (ref: physicians: AOR = 0.39, 95% CI [0.18–0.58], $p = 0.02$) and lab technicians (ref: physicians: AOR = 0.31, 95% CI [0.11–0.53], $p = 0.02$) had a significant lower practice score. The test results revealed a weak positive correlation of knowledge with the attitude scores (rho = 0.249, $p = 0.001$), and a moderately strong positive correlation was found between attitude and practice scores (rho = 0.432, $p = 0.001$).

**Conclusion.** This study suggests that a regular training program for the HCWs on BMW management is necessary through symposiums, role play, interactive lectures, and other feasible training methods. Furthermore, a multicentric prospective exploratory study is

Corresponding author
Ashokkumar Thirunavukkarasu,
ashokkumar@ju.edu.sa

to be conducted in other regions of the KSA to understand the region-specific training needs of HCWs.

## INTRODUCTION

Biomedical waste (BMW) is defined as "any type of waste that is generated during diagnosis, treatment, or research in humans and/or animals" (*Singh et al., 2001*). As per the World Health Organization (WHO), around 85% of the total amount of waste generated during healthcare activities is non-hazardous waste similar to domestic waste, and only 15% is considered hazardous waste that includes various forms of waste such as human anatomical parts, blood and tissues, radiation waste, chemo-toxic drugs, and broken medical equipment (*WHO, 2021*). BMW is commonly generated in hospitals, primary health care facilities, medical colleges, research centers, and veterinary hospitals (*Pandey et al., 2016*; *Meleko, Tesfaye & Henok, 2018*).

Advancement in medical care and the introduction of more advanced equipment has dramatically increased the amount of waste generated per patient in healthcare facilities worldwide. Globally, a large amount of health care waste generated everyday ranges from 0.9 kg/bed/day to 3 kg/bed/day. In the Kingdom of Saudi Arabia (KSA), the amount of health care generation is reported as 1.66 kg/bed/day (*Alagha, Alomari & Jarrah, 2018*). The WHO stated that around 16 billion injections are given every year for treatment or immunization. Unfortunately, not all used needles and syringes are disposed of as per the legislation and safety norms, leading to the risk of needle prick injury and acquiring the related infection. This also provides opportunities for reuse (*WHO Injection Safety Geneva, 2022*). The COVID-19 pandemic worsens the existing healthcare waste burden as healthcare waste management during the COVID-19 pandemic has faced several challenges due to increased production of infectious waste, interruption of recycling strategy, and inadequate resources to handle increased waste production (*Dehal, Vaidya & Kumar, 2022*; *Kothari et al., 2021*; *Rahman et al., 2020*; *Das et al., 2021*).

Health care workers (HCWs) involved in day-to-day care to the patients and other health care activities play a significant role in BMW management (*Wafula, Musiime & Oporia, 2019*; *Letho et al., 2021*). The WHO classifies these health workers into several categories: general and specialist medical practitioners, nurses and midwifery professionals, complementary medicine practitioners, pharmacists, physiotherapists, etc. (*WHO, 2021*). The WHO stated some of the causes for the failure of BMW management, namely, lack of awareness about the health hazards related to BMW, inadequate trained HCWs in BMW management, lack of legislation and rules regarding waste management systems, inadequate human and monetary resources (*WHO Healthcare Waste Geneva, 2021*).

*Deress et al. (2018)* conducted a study in 2018 that assessed the KAP of healthcare professionals of northwest Ethiopia towards BMW management and associated factors

for good KAP scores. They found that 56.8%, 66.2%, and 77.4% of the HCWs had good knowledge, attitude, and practice scores, respectively. Regarding the associated factors, those who had higher level qualifications and the availability of color-coded bins were more likely to have favorable KAP scores. Another survey conducted by *Jalal et al. (2021)* during COVID-19 times that evaluated the KAP toward BMW management among healthcare professionals stated that nearly a quarter of healthcare professionals had poor knowledge regarding BMW management. The mean knowledge scores were significantly higher among nurses and physicians than others.

In the KSA, infection control and BMW management are implemented uniformly in all healthcare facilities according to ministry of health (MOH), KSA and Gulf Cooperation Council Centre for Infection Control policies (*NGHA, 2022*). The infection prevention and control departments have primary responsibilities to train HCWs. All national HCWs seeking jobs in healthcare facilities must clear the Saudi Commission for Health Specialties test to qualify for health sector jobs. However, there are no specific requirements for separate and compulsory BMW management and infection control training to apply for health sector jobs and contract renewal. Assessment of knowledge, attitude, and practices (KAP) towards BMW among the HCWs are essential in planning the training program for them towards BMW management to the extent they need (*Al Balushi et al., 2018*; *Reddy & Al Shammari, 2017*; *Olaifa, Govender & Ross, 2018*) so that they will dispose of the BMW according to the regulation implemented in their hospitals; this will eventually decrease the health hazards that will arise from BMW (*Alqahtani et al., 2019*; *Aliyu et al., 2017*). To the best of the authors' knowledge, there is limited data available in northern Saudi Arabia on this topic. Therefore, the present study was planned to assess knowledge, attitude, and practice in the management of biomedical waste among the HCWs of public healthcare facilities in Aljouf province, KSA, to determine the influencing factors on knowledge, attitude, and practices among them and to identify the correlation between the scores of knowledge, attitude, and practices.

## PARTICIPANTS & METHODS

### Study design and setting

The present cross-sectional survey was conducted from December 2021 to February 2022 among the HCWs from different healthcare facilities in Aljouf province, Saudi Arabia. The Aljouf province is located in the northern regions of the KSA. The present study included the HCWs working in the healthcare facilities under the ministry of health (MOH) for a minimum of one year. Currently, in the KSA, healthcare is provided through four levels, namely, primary health centers (PHC), general hospitals, specialty hospitals, and medical cities. There are 62 primary health centers, 13 general hospitals, and two specialty hospitals in the Aljouf province under the MOH, KSA. The HCWs from the infection control department and those not willing to participate were excluded from the study. Also, those on vacation were excluded from the sampling frame when selecting sample participants.

## Sample size and sampling method

We calculated the required sample for this study based on Cochran's sample size estimation equation (n = z2pq/e2). In this formula, n is the number of participants required, $z = 1.96$ in 95% confidence interval, $p$ = expected proportion, $q = 1$-p, and e is 5% of margin of error. Since previous studies reported different prevalence on this subject, we have taken 50% as the expected proportion to obtain the maximum sample size. Applying the values mentioned above in Cochran's equation, the estimated sample size for the present study was 384.

The research team applied the multistage probability proportional sampling (PPS) method to select the required number of HCWs in each category. Firstly, we have included all the 62 PHCs from the Aljouf province. Then a general hospital (from 13 hospitals) and a tertiary care hospital (from two hospitals) in the Aljouf region were selected using the lot method. The required number of participants was selected from each type of healthcare facility based on the total HCWs registered in the facilities. After obtaining approval from selected healthcare facilities, we have arranged the total HCWs according to the assigned numbers in the ascending order for each type of healthcare facility. Finally, we have applied a systematic random sampling method according to the allotted number to select the participants from each healthcare facility.

## Data collection methods

The present study data collection began after obtaining the required ethical clearance from the regional ethics committee, Aljouf region (Local research ethics committee, Qurayat Health Affairs, Ministry of Health, KSA: Approval number-116), and concerned healthcare facilities authorities. The research team communicated with the selected participants for their availability to collect data. Data collectors made three attempts to contact the selected participants in one month. A selected participant who could not be reached despite three attempts in a month and who was unwilling to participate was considered nonrespondent. The next HCW of the same category was invited to participate in the survey in case of nonrespondent.

## Survey questionnaire

After briefing the study objectives and obtaining written informed consent from the participants of the present study, we conducted the data collection process using a self-administered, standard, validated questionnaire adapted from a previously published study (*Jalal et al., 2021*). The research team obtained written permission from the corresponding author of the study by Jalal SM et al. to use and adapt/modify for local settings. After receiving the data collection tool (with the necessary permission to adapt and modify) from the reference study author, the present research team modified it to reduce the duration of the survey to improve the validity and reliability as suggested by existing literatures (*Story & Tait, 2019*; *Kost & de Rosa, 2018*). We have followed the following steps to test the validity and reliability of the adapted questionnaire before proceeding to collect data. Firstly, the adapted tool is reviewed by the community medicine and infection control department experts through a focus group discussion. Secondly, we have conducted a pilot study

among thirty different category HCWs in the local settings. All pilot study respondents acknowledged that the data collection form was simple, clear, and easy to understand. Furthermore, there was no missing data in all thirty completed forms. Cronbach's alpha ($\alpha$) values for the knowledge, attitude, and practice scores were 0.75, 0.83, and 0.86, which exhibited good internal consistency in the current form of the data collection tool. A google form was created and given to the selected participants in the personal digital device of the data collector. However, only the principal investigator had the authorization to access and download the spreadsheet. The data collection form consisted of two sections. The first section inquired about the participants' background details, including age, gender, education status, marital status, nationality, HCW category, and duration of work experience. In our study, HCWs were grouped into five categories, namely, medical practitioners, nursing and midwifery professionals, laboratory technicians, pharmacists, and others (this includes remaining all categories mentioned in the classification of WHO) (*WHO, 2021*). We categorized the continuous data (age and duration of work experience) as per the class interval. Since the Ph.D./Fellowship in education status category is less than 30, as a rule of thumb, we combined it with masters and regrouped it into a master's and above. A similar rule we have applied for the marital status for grouping. The second section consisted of ten questions in knowledge, attitude, and practice categories. In the knowledge section, we inquired about participants' knowledge regarding different forms and sources of BMW, disposal methods, and hazards due to improper BMW management. Correct answers were given as one mark in the knowledge category, and wrong answers were given zero marks. In both attitude and practice categories, the participants responded on a 5-point Likert scale as "strongly agree," "agree," "neutral," "disagree," and "strongly disagree," which were given scores 5, 4, 3, 2, and 1, respectively. Overall knowledge, attitude, and practice scores were interpreted as high (score 80% and above), medium (60 to 79%), and low (less than 60%).

## Statistical analysis

The spreadsheet (excel) is exported to the statistical package for social science (SPSS) version 20 and coded as per the predefined coding sheet for further analysis. The present study's descriptive statistics are presented as frequency (n) and proportion (%). In Saudi Arabia, the HCWs are expected to have excellent/good knowledge ($\geq$ 80%) in infection control and other common public health activities as they are the first in line in managing the BMW. Our categorization is in accordance with the original Bloom's cut-off point for assessing KAP and is supported by previous studies conducted among healthcare workers in the KSA and other parts of the world (*Mohammed Basheeruddin Asdaq et al., 2021*; *Feleke, Wale & Yirsaw, 2021*; *Abalkhail et al., 2021*). Additionally, we combined low and medium scores as a single category for further analysis, and previous studies among the HCWs strongly support our categorization (*Abalkhail et al., 2021*; *Thirunavukkarasu et al., 2021*; *Ukwenya et al., 2021*). The association of the subscales' categories and sociodemographic characteristics was assessed by Pearson's chi-square test. Furthermore, we performed binomial logistic regression analysis to find the predictors for the subscales of KAP. In this technique, we adjusted the covariables of the study to get an adjusted odds ratio (AOR)

The Kolmogorov–Smirnov normality test (KS) of the knowledge, attitude, and practice data did not meet the normality assumption. Therefore, we have executed Spearman's correlation test to find the strength and direction of correlation (*rho*) between knowledge, attitude, and practice scores (*de Winter, Gosling & Potter, 2016*). All statistical tests used in this study were two-tailed and a *p*-value less than 0.05 was established as a statistically significant value.

## RESULTS

Table 1 depicts the sociodemographic characteristics of the respondents. Of the 384 participants, the majority (59.9%) respondents were males, Saudi nationals (71.1%), married (46.1%), highest education qualification as bachelor's degree (60.4%), and had 5 to 10 years of work experience in the healthcare settings (34.6%). Of the sample studied, 35.9% of the HCWs were nurses and midwiferies, 37.0% were working at general hospitals, and the mean ($\pm$SD) of the age of the participants was $34.35 \pm 9.5$ years.

Figure 1 presents the categorization (low/medium/high) of KAP scores. Of the 384 respondents, high knowledge, attitude, and practice scores were found in 47.1%, 65.1%, and 49.5% of the HCWs, respectively.

The cross-tabulation between knowledge score categories and sociodemographic characteristics found a significant association between age groups ($p = 0.001$), nationality ($p = 0.001$), education status ($p = 0.018$), and work experience duration ($p = 0.01$). Of the studied participants, high knowledge was noticed among the HCWs aged 40 years and above (56.1%), who had higher education qualifications (65.5%), and work experience of more than 10 years (58.1%) (Table 2).

The cross-tabulation between attitude score categories and sociodemographic characteristics is presented in Table 3. Of the 384 samples analyzed, a significant association was found among health care working settings ($p = 0.005$) and work experience ($p = 0.001$). No other sociodemographic characteristics, including age, gender, nationality, marital status, education and HCWs category had a significant association with the attitude score categories.

Cross-tabulation between practice score categories and sociodemographic characteristics found a significant association between age groups ($p = 0.001$), gender ($p = 0.008$), nationality ($p = 0.001$), marital status ($p = 0.020$), and duration of work experience ($p = 0.001$) (Table 4).

Binomial logistic regression analysis on KAP subscales (low/medium vs high) and its association with participants sociodemographic characteristics are presented in Table 5. The present study found that knowledge score was significantly higher among the age group from 30 to 39 years (ref: age less than 30 years: AOR = 2.25, 95% CI [1.05–4.85], $p = 0.04$), non- Saudi nationals (ref: Saudi: AOR = 2.84, 95% CI [1.63–4.94], $p < 0.001$), and those with higher education (ref: diploma holder: AOR for bachelors = 1.98, 95% CI [1.06−3.68], $p = 0.04$ and AOR for masters and above = 3.78, 95% CI [1.59−8.97], $p = 0.03$). The attitude score towards BMW management was significantly lower among the HCWs working in tertiary care settings (ref: PHC: AOR = 0.38, 95% CI [0.12–0.69],

**Table 1** Socio-demographic characteristics of the healthcare workers (HCWs) ($n = 384$).

| Variables | Frequency (n) | Proportion (%) |
|---|---|---|
| Age (in years) | | |
| Mean $\pm$ SD | 34.35 $\pm$ 9.5 | |
| Less than 30 | 144 | 37.5 |
| 30 to 39 | 133 | 34.6 |
| 40 and above | 107 | 27.9 |
| Gender | | |
| Male | 230 | 59.9 |
| Female | 154 | 40.1 |
| Nationality | | |
| Saudi | 273 | 71.1 |
| Non-Saudi | 111 | 28.9 |
| Marital status | | |
| Single | 167 | 43.5 |
| Married | 177 | 46.1 |
| Divorced/Widowed | 40 | 10.4 |
| Education | | |
| Diploma | 82 | 21.4 |
| Bachelors | 232 | 60.4 |
| Masters and above | 70 | 18.2 |
| HCW category | | |
| Physicians | 116 | 30.2 |
| Nursing and midwife | 138 | 35.9 |
| Pharmacist | 48 | 12.5 |
| Lab technicians | 31 | 8.1 |
| Other categories[a] | 51 | 13.3 |
| Work settings | | |
| Primary health centers (PHC) | 127 | 33.1 |
| General hospital | 142 | 37.0 |
| Tertiary care hospital | 115 | 39.9 |
| Work experience (in years) | | |
| Less than 5 | 127 | 33.1 |
| 5 to 10 | 133 | 34.6 |
| More than 10 | 124 | 32.3 |

**Notes.**

[a] Physiotherapist, Dentists, community health workers, radiology technicians, and Occupational health and hygiene personnel.

$p = 0.01$). Regarding the practice score, the male categories had a significantly higher score (ref: female: AOR $= 1.82$, 95% CI [1.19–2.99], $p = 0.02$), while pharmacist (ref: physicians: AOR $= 0.39$, 95% CI [0.18–0.58], $p = 0.02$) and lab technicians (ref: physicians: AOR $= 0.31$, 95% CI [0.11–0.53], $p = 0.02$) had a significant lower practice score.

The present study data did not meet the normality assumption criteria. Therefore, we have executed Spearman's correlation test. The test results revealed a weak positive correlation of knowledge with the attitude ($rho = 0.249$, $p = 0.001$) and practice scores

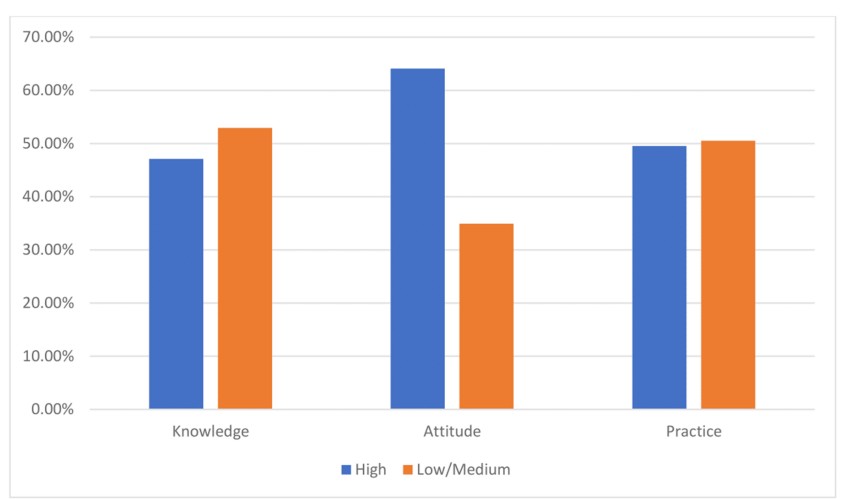

**Figure 1** Distribution of knowledge, attitude, and practice (KAP) scores categories among the participants ($n = 384$).

($rho = 0.104$, $p = 0.042$). Also, a moderately strong positive correlation was found between attitude and practice scores ($rho = 0.470$, $p = 0.001$) (Table 6).

## DISCUSSION

Improper handling of medical waste generated at health care facilities may pose a serious threat to the HCWs, common people and the surrounding environment, asserted by the WHO, UN, and CDC (*WHO, 2021*; *CDC, 2021*). Since the HCWs play an important role in regulated biomedical waste disposal, it is important to have a high level of awareness. This emphasizes the importance of evaluating the knowledge, attitude, and practice of HCWs with respect to BMW management and the factors that influence them.

Knowledge is an essential resource in health science education, and inadequate knowledge may lead to improper application of knowledge that may be detrimental to any healthcare organization (*Karimi, Hosseinian & Ahanchian, 2014*; *Shahmoradi, Safadari & Jimma, 2017*). The present study results revealed that less than half (47.1%) of the participants had high knowledge of medical waste management generated at their work settings. Similar to our study findings, a recent survey conducted by Jalal SM in the Al-Hasa region of the KSA also found that only 41% of the healthcare professionals had excellent knowledge of biomedical waste disposal (*Jalal et al., 2021*). Interestingly, surveys conducted in some other countries also reported that a low proportion of the HCWs had favorable knowledge of healthcare handling and disposal (*Deress et al., 2018*; *Olaifa, Govender & Ross, 2018*; *Woromogo et al., 2020*; *Krithiga et al., 2021*). In contrast to our study, a study conducted by *Reddy & Al Shammari (2017)* in the Hail region of the KSA and *Akkajit, Romin & Assawadithalerd (2020)* in Thailand stated that a higher proportion of healthcare professionals had good knowledge. The possible difference between our study and the latter studies could be the inclusion of healthcare facilities. The present study included multiple

**Table 2  Cross-tabulation between low/medium, high knowledge score categories and sociodemographic characteristics ($n = 384$).**

| Variables | Low/Medium n (%) | High n (%) | p - value |
|---|---|---|---|
| Age (in years) | | | |
| Less than 30 | 100 (69.4) | 44 (30.6) | |
| 30 to 39 | 56 (42.1) | 77 (57.9) | 0.001* |
| 40 and above | 47 (43.9) | 60 (56.1) | |
| Gender | | | |
| Male | 121 (52.6) | 109 (47.4) | 0.917 |
| Female | 82 (53.2) | 72 (46.8) | |
| Nationality | | | |
| Saudi | 165 (60.4) | 108 (39.6) | 0.001* |
| Non-Saudi | 38 (34.2) | 73 (65.8) | |
| Marital status | | | |
| Single | 106(63.5) | 61 (36.5) | |
| Married | 75 (42.4) | 102 (57.6) | 0.090 |
| Divorced/Widowed | 22 (55.5) | 18 (45) | |
| Education | | | |
| Diploma | 57 (69.5) | 25 (30.5) | |
| Bachelors | 122 (52.6) | 110 (47.4) | 0.018* |
| Masters and above | 24 (34.3) | 46 (65.7) | |
| HCW category | | | |
| Physicians | 57 (49.1) | 59 (50.9) | |
| Nursing and midwife | 70 (50.7) | 68 (49.3) | 0.307 |
| Pharmacist | 24 (50.0) | 24 (50.0) | |
| Lab technicians | 19 (61.3) | 12 (38.7) | |
| Other categories | 33 (64.7) | 18 (35.3) | |
| Work settings | | | |
| Primary health centers (PHC) | 70 (55.1) | 57 (44.9) | |
| General hospital | 80 (56.3) | 62 (43.7) | 0.216 |
| Tertiary care hospital | 53 (46.1) | 62 (53.9 | |
| Work experience (in years) | | | |
| Less than 5 | 85 (66.9) | 42 (33.1) | |
| 5 to 10 | 66 (49.6) | 67 (50.4) | 0.001* |
| More than 10 | 52 (41.9) | 72 (58.1) | |

**Notes.**
*Significant $p$-value ($<0.05$) from Chi-square test.

healthcare facilities (PHCs, general and tertiary hospitals), while later included outpatient clinics and PHCs.

Our study found a significant association between knowledge category and age groups (ref: age less than 30 years: AOR = 2.25, 95% CI [1.05–4.85], $p = 0.04$), nationality(ref: Saudi: AOR = 2.84, 95% CI [1.63–4.94], $p < 0.001$), and education status (ref: diploma holder: AOR = 3.78, 95% CI [1.59–8.97], $p = 0.03$) Other surveys conducted in the KSA, and other countries support the present study results. *Jalal et al. (2021)* conducted

**Table 3** Cross-tabulation between low/medium, high attitude score categories and sociodemographic characteristics ($n = 384$).

| Variables | Low/Medium n (%) | High n (%) | p - value |
|---|---|---|---|
| Age (in years) | | | |
| Less than 30 | 40 (27.8) | 104 (72.2) | |
| 30 to 39 | 50 (37.6) | 83 (62.4) | 0.065 |
| 40 and above | 44 (41.1) | 63 (58.9) | |
| Gender | | | |
| Male | 76 (33.0) | 154 (67.0) | 0.352 |
| Female | 58 (37.7) | 96 (62.3) | |
| Nationality | | | |
| Saudi | 93 (34.1) | 180 (65.9) | 0.593 |
| Non-Saudi | 41 (36.9) | 70 (63.1) | |
| Marital status | | | |
| Single | 50 (29.9) | 117 (70.1) | 0.132 |
| Married | 66 (37.3) | 111 (62.7) | |
| Divorced/Widowed | 18 (45.0) | 22 (55.0) | |
| Education | | | |
| Diploma | 32 (39.0) | 50 (61.0) | |
| Bachelors | 80 (34.5) | 152 (65.5) | 0.606 |
| Masters and above | 22 (31.4) | 48 (68.6) | |
| HCW category | | | |
| Physicians | 39 (33.6) | 77 (66.4) | 0.535 |
| Nursing and midwife | 43 (31.2) | 95 (68.8) | |
| Pharmacist | 17 (35.4) | 31 (64.6) | |
| Lab technicians | 13 (41.9) | 18 (58.1) | |
| Other categories | 22 (43.1) | 29 (56.9) | |
| Work settings | | | |
| Primary health centers (PHC) | 32 (25.2) | 95 (74.8) | 0.005[*] |
| General hospital | 50 (35.2) | 92 (64.8) | |
| Tertiary care hospital | 52 (45.1) | 63 (54.8) | |
| Work experience (in years) | | | |
| Less than 5 | 31 (24.4) | 96 (75.6) | |
| 5 to 10 | 64 (48.1) | 69 (51.9) | 0.001[*] |
| More than 10 | 39 (31.5) | 85 (68.5) | |

**Notes.**
*Significant $p$-value ($<0.05$) from Chi-square test.

a survey in 2021 which also revealed a statistically significant association between the excellent knowledge category and the level of education qualification, genders, and work experience. However, the present study did not find a significant association between gender and knowledge category. Educational qualification is one of the important factors that influence the ability to have high knowledge. The results of the current study revealed that HCWs who had a higher level of education had a significantly higher level of knowledge than diploma holders (ref: diploma holder: AOR = 3.78, 95% CI [1.59−8.97], $p = 0.03$).

**Table 4  Cross-tabulation between low/medium, high practice score categories and sociodemographic characteristics ($n = 384$).**

| Variables | Low/Medium n (%) | High n (%) | p - value |
|---|---|---|---|
| **Age (in years)** | | | |
| Less than 30 | 52 (36.1) | 92 (63.9) | |
| 30 to 39 | 77 (57.9) | 56 (42.1) | 0.001[*] |
| 40 and above | 65 (60.7) | 42 (39.3) | |
| **Gender** | | | |
| Male | 97 (42.2) | 133 (57.8) | 0.008[*] |
| Female | 97 (63.0) | 57 (37.0) | |
| **Nationality** | | | 0.001[*] |
| Saudi | 110 (40.3) | 163 (59.7) | |
| Non-Saudi | 84 (75.7) | 27 (24.3) | |
| **Marital status** | | | |
| Single | 67 (40.1) | 100 (59.9) | |
| Married | 104 (58.8) | 73 (41.2) | 0.020[*] |
| Divorced/Widowed | 23 (57.5) | 17 (42.5) | |
| **Education** | | | |
| Diploma | 32 (39.0) | 50 (61.0) | 0.092 |
| Bachelors | 129 (55.6) | 103 (44.4) | |
| Masters and above | 33 (47.1) | 37 (52.9) | |
| **HCW category** | | | |
| Physicians | 47 (40.5) | 69 (59.5) | 0.071 |
| Nursing and midwife | 76 (55.1) | 62 (44.9) | |
| Pharmacist | 29 (60.4) | 19 (39.6) | |
| Lab technicians | 18 (58.1) | 13 (41.9) | |
| Other categories | 24 (47.1) | 27(52.9) | |
| **Work settings** | | | |
| Primary health centers (PHC) | 71 (55.9) | 56 (44.1) | 0.323 |
| General hospital | 67 (47.2) | 75 (52.8) | |
| Tertiary care hospital | 56 (48.7) | 59 (51.3) | |
| **Work experience (in years)** | | | |
| Less than 5 | 44 (34.6) | 83 (65.4) | 0.001[*] |
| 5 to 10 | 81 (60.9) | 52 (39.1) | |
| More than 10 | 69 (55.6) | 55 (44.1) | |

**Notes.**
  *Significant p-value (<0.05) from Chi-square test.

Similar to the current study findings, other surveys conducted by *Deress et al. (2018)*, *Reddy & Al Shammari (2017)* and *Dixit et al. (2021)* also revealed a positive association among highly qualified healthcare professionals. It is worth mentioning again that all national HCWs working in healthcare facilities had to clear the Saudi Commission for Health Specialties test to get eligible for health sector jobs. However, there are no specific requirements for separate and compulsory BMW management and infection control training to apply for health sector job and contract renewal. A study done by *Al-Ahmari,*

**Table 5 Binomial logistic regression analysis on KAP subscales (low/medium vs high) and its association with participants sociodemographic characteristics ($n = 384$).**

| Variables | Total | Knowledge | | | | Attitude | | | | Practice | | | |
|---|---|---|---|---|---|---|---|---|---|---|---|---|---|
| | | Low/ Medium n(%) | High n(%) | Adjusted OR (95% CI)** | p-value | Low/ Medium n(%) | High n(%) | Adjusted OR (95% CI)** | p-value | Low/ Medium n(%) | High n(%) | Adjusted OR (95% CI)** | p-value |
| Age (in years) | | | | | | | | | | | | | |
| Less than 30 | 144 | 100 (69.4) | 44 (30.6) | Ref | | 40 (27.8) | 104 (72.2) | Ref | | 52 (36.1) | 92 (63.9) | Ref | |
| 30 to 39 | 133 | 56 (42.1) | 77 (57.9) | 2.25 (1.05–4.85) | 0.04* | 50 (37.6) | 83 (62.4) | 0.99 (0.46–2.13) | 0.98 | 77 (57.9) | 56 (42.1) | 0.70 (0.33–1.52) | 0.37 |
| 40 and above | 107 | 47 (43.9) | 60 (56.1) | 1.38 (0.52–3.67) | 0.52 | 44 (41.1) | 63 (58.9) | 0.39 (0.14–1.08) | 0.07 | 65 (60.7) | 42 (39.3) | 0.50 (0.18–1.40) | 0.19 |
| Gender | | | | | | | | | | | | | |
| Female | 154 | 82 (53.2) | 72 (46.8) | Ref | | 58 (37.7) | 96 (62.3) | Ref | | 97 (63.0) | 57 (37.0) | Ref | |
| Male | 230 | 121 (52.6) | 109(47.4) | 1.24 (0.76–2.05) | 0.38 | 76 (33.0) | 154 (67.0) | 1.05 (0.64–1.74) | 0.84 | 97 (42.2) | 133 (57.8) | 1.82 (1.10–2.99) | 0.02* |
| Nationality | | | | | | | | | | | | | |
| Saudi | 273 | 165 (60.4) | 108 (39.6) | Ref | | 93 (34.1) | 180 (65.9) | Ref | | 110 (40.3) | 163 (59.7) | Ref | |
| Non-Saudi | 111 | 38 (34.2) | 73 (65.8) | 2.84 (1.63–4.94) | 0.00* | 41 (36.9) | 70 (63.1) | 1.23 (0.71–2.14) | 0.47 | 84 (75.7) | 27 (24.3) | 0.30 (0.17–1.52) | 0.10 |
| Marital status | | | | | | | | | | | | | |
| Single | 167 | 106 (63.5) | 61(36.5) | Ref | 50 (29.9) | 117 (70.1) | Ref | 67 (40.1) | 100 (59.9) | Ref | | | |
| Married | 177 | 75 (42.4) | 102 (57.6) | 1.60 (0.95–2.69) | 0.07 | 66 (37.3) | 111 (62.7) | 0.79 (0.46–1.35) | 0.39 | 104 (58.8) | 73 (41.2) | 0.60 (0.35–1.03) | 0.06 |
| Divorced/Widowed | 40 | 22 (55.0) | 18 (45.0) | 0.87 (0.38–1.99) | 18 (45.0) | 22 (55.0) | 0.58 (0.26–1.32) | 0.19 | 23 (57.5) | 17 (42.5) | 0.82(0.35–1.90) | 0.64 | |
| Education | | | | | | | | | | | | | |
| Diploma | 82 | 57 (69.5) | 25 (30.5) | Ref | | 32 (39.0) | 50 (61.0) | Ref | | 32 (39.0) | 50 (61.0) | Ref | |
| Bachelors | 232 | 122 (52.6) | 110 (47.4) | 1.98 (1.06–3.68) | 0.04* | 80 (34.5) | 152 (65.5) | 1.23 (0.66–2.27) | 0.52 | 129 (55.6) | 103 (44.4) | 0.41 (0.22–0.78) | 0.01* |
| Masters and above | 70 | 24 (34.3) | 46 (65.7) | 3.78 (1.59–8.97) | 0.03* | 22 (31.4) | 48 (68.6) | 2.13 (0.89–5.09) | 0.09 | 33 (47.1) | 37 (42.9) | 0.47 (0.20–1.14) | 0.09 |
| HCW category | | | | | | | | | | | | | |
| Physicians | 116 | 57 (49.1) | 59 (50.9) | Ref | | 39 (33.6) | 77 (66.4) | Ref | | 47 (40.5) | 69 (59.5) | Ref | |
| Nursing and midwife | 138 | 70 (50.7) | 68 (49.3) | 1.35 (0.73–2.49) | 0.34 | 43 (31.2) | 95 (68.8) | 1.16 (0.62–2.16) | 0.65 | 76 (55.1) | 62 (44.9) | 0.53 (0.28–1.01) | 0.05 |
| Pharmacist | 48 | 24 (50.0) | 24 (50.0) | 1.30 (0.59–2.85) | 0.51 | 17 (35.4) | 31 (64.6) | 1.28 (0.58–2.82) | 0.55 | 29 (60.4) | 19 (39.6) | 0.39 (0.18–0.58) | 0.02* |
| Lab technicians | 31 | 19 (61.3) | 12 (38.7) | 1.12 (0.43–2.88) | 0.82 | 13 (41.9) | 18 (58.1) | 1.09 (0.43–2.76) | 0.86 | 18 (58.1) | 13 (41.9) | 0.31 (0.11–0.53) | 0.02* |
| Other categories | 51 | 33 (64.7) | 18 (35.3) | 0.88 (0.68–2.02) | 0.76 | 22 (43.1) | 29 (56.9) | 0.96 (0.42–2.76) | 0.92 | 24 (47.1) | 27 (52.9) | 0.60 (0.25–1.42) | 0.25 |
| Work settings | | | | | | | | | | | | | |
| PHC | 127 | 70 (55.1) | 57 (44.9) | Ref | | 32 (25.2) | 95 (74.8) | Ref | | 71 (55.9) | 56 (44.1) | Ref | |
| General hospital | 142 | 80 (56.3) | 62 (43.7) | 0.98 (0.58–1.67) | 0.94 | 50 (35.2) | 92 (64.8) | 0.61 (0.35–1.07) | 0.08 | 67 (47.2) | 75 (52.8) | 1.63 (0.94–2.83) | 0.08 |
| Tertiary care hospital | 115 | 53 (46.1) | 62 (53.9) | 1.30 (0.74–2.29) | 0.37 | 52 (45.2) | 63 (54.8) | 0.38 (0.12–0.69) | 0.01* | 56 (48.7) | 59 (51.3) | 1.67 (0.92–3.02) | 0.09 |
| Work experience (years) | | | | | | | | | | | | | |
| Less than 5 | 127 | 85 (66.9) | 42(33.1) | Ref | | 31 (24.4) | 96 (75.6) | Ref | | 44 (34.6) | 83 (65.4) | Ref | |
| 5 to 10 | 133 | 66 (49.6) | 67 (50.4) | 0.84 (0.39–1.77) | 0.64 | 64 (48.1) | 69 (51.9) | 0.40 (1.91–0.85) | 0.20 | 81 (60.9) | 52 (39.1) | 0.66 (0.32–1.37) | 0.26 |
| More than 10 | 124 | 52 (41.9) | 72 (58.1) | 0.99 (0.39–2.57) | 0.99 | 39 (31.5) | 85 (68.3) | 1.43 (0.52–3.88) | 0.49 | 69 (55.6) | 55 (44.4) | 1.14 (0.43–2.99) | 0.79 |

**Notes.**

*Significant association ($p < 0.05$).

**Variable adjusted in enter method: age, gender, nationality, marital status, education status, HCWs category, work setting, and work experience.

**Table 6  Spearman's correlation analysis between KAP scores.**

| Variable | rho[*] / p - value |
|---|---|
| Knowledge–Attitude | .249/0.002[**] |
| Knowledge–Practice | .104/0.042[**] |
| Attitude–Practice | 0.470/0.001[**] |

**Notes.**
[*]Spearman's rank correlation coefficient.
[**]Significant at 0.05 level (2-tailed).

*Alkhaldi & Al-Asmari (2021)* reported that the majority of the primary care professionals did not receive sufficient infection control training programs, which affects their knowledge significantly. The Saudi government initiated Saudization for health sector jobs and young Saudi graduates are entering the job market. This could be the possible reason for the significant association between knowledge scores with the nationality and age group (*Elsheikh et al., 2018*; *Al-Hanawi, Khan & Al-Borie, 2019*).

A positive attitude will guide the HCWs to follow the standards, protocols, and evidence-based practices established by the healthcare organization (*Mariano et al., 2018*; *Sayankar, 2015*). This study found that nearly two-thirds of the participants had a high attitude towards biomedical waste disposal (65.1%). Using binomial logistic regression analysis, a significant association with attitude was found among different work settings (ref: PHC: AOR = 0.38, 95% CI [0.12–0.69], $p = 0.01$), and no other sociodemographic variables were significantly associated with attitude. Identical to our study findings, a study by (*Dalui, Banerjee & Roy, 2021*) also found that a high proportion of healthcare providers had an excellent attitude towards BMW management. Another study conducted in Cairo, Egypt, reported that the duration of work experience was not significantly associated with attitude (*Hakim, Mohsen & I, 2014*). Interestingly, some studies found that doctors had a higher positive attitude towards healthcare waste disposal than nurses and other HCWs (*Reddy & Al Shammari, 2017*; *Hakim, Mohsen & I, 2014*; *Basavaraj, Shashibhushan & Sreedevi, 2021*). These huge variations in the results among different studies could be due to the variations in data collection tools, survey settings, and cultural variations.

The present study results revealed that only half of the participants had high scores in practice. Our study found a positive association with practice scores were found among male gender (ref: female: AOR = 1.82, 95% CI [1.19–2.99], $p = 0.02$), and HCW category (ref: physicians: AOR = 0.39, 95% CI [0.18–0.58], $p = 0.02$). Similar to our study, *Reddy & Al Shammari (2017)* also reported that only 50% of the HCWs had excellent practice scores. In contrast to the present study results, a survey conducted in Ethiopia reported that a higher proportion of HCWs had a satisfactory practice score (*Deress et al., 2018*). In contrast to the current study findings, some other surveys found a positive association of duration of work experience and older age with the practice scores (*Reddy & Al Shammari, 2017*; *Akkajit, Romin & Assawadithalerd, 2020*; *Hakim, Mohsen & I, 2014*). Similar to this study results, some studies found a positive association with the type of HCW (*Dalui, Banerjee & Roy, 2021*; *Rao et al., 2018*). The present study's results revealed that BMW management practices were not significantly associated with the marital status of the

participants. Similarly, a study conducted by *Desta et al. (2018)* did not find an association between marital status and good practice. Interestingly, a survey conducted in the KSA in 2021 on assessing KAP among the HCWs on the COVID-19 prevention found a significant association between marital status and appropriate practices ($p = 0.024$) (*Almohammed et al., 2021*).

The Spearman's rank correlation test results revealed a weak positive correlation of knowledge with the attitude scores ($rho = 0.249$, $p = 0.001$), and a moderately strong positive correlation was found between attitude and practice scores ($rho = 0.432$, $p = 0.001$). These findings reassert the importance of association between KAP for the proper BMW management. Furthermore, our results conclude that the HCW's favorable knowledge led to positive attitude and proper practice. Our study results are supported by several studies that assessed KAP towards healthcare waste management in different countries (*Reddy & Al Shammari, 2017*; *Woromogo et al., 2020*; *Akkajit, Romin & Assawadithalerd, 2020*).

Even though the present study was conducted with the proper methodology and adequate sample size among different HCWs working in multiple healthcare facilities, certain limitations are to be noted on reading the results of this survey. Firstly, we assessed only the association through this cross-sectional survey, not the causation and direction. Secondly, the possible bias associated with the self-reported data could influence the results of this survey. Finally, this survey was conducted in the northern region of the KSA, and therefore, the findings cannot be generalized to the other areas of the KSA and other countries in the Middle East.

## CONCLUSIONS

The present study assessed KAP towards BMW among healthcare providers working in different healthcare facilities using a standard and validated tool. Our study revealed that less than half of the participants had insufficient knowledge and practice scores, while one-third had low and medium attitude scores. Furthermore, our results conclude that the HCW's good knowledge may lead to a positive attitude and proper practice. The findings of this study suggest that a regular training program for the HCWs on BMW management is necessary through symposiums, role-play, interactive lectures, and other feasible training methods. These training programs can be focused and targeted oriented to the HCWs category with low and medium scores in KAP. Finally, a multicentric prospective exploratory study is to be conducted in other regions of the KSA to understand the region-specific training needs of the HCWs.

## ACKNOWLEDGEMENTS

The research team wish to thank all healthcare workers for their participation in this study. We also extend our thanks to the healthcare facilities for facilitating data collection in their healthcare settings. Finally, we wish to thank Dr. Bashayer Farhan ALruwailli, Assistant Professor of Family Medicine, Jouf University for her input for the article during revision time.

### Funding

This work was funded by the Deanship of Scientific Research at Jouf University under grant number (DSR-2021-01-03144). The funders had no role in study design, data collection and analysis, decision to publish, or preparation of the manuscript.

### Grant Disclosures

The following grant information was disclosed by the authors:
the Deanship of Scientific Research at Jouf University: DSR-2021-01-03144.

### Competing Interests

The authors declare there are no competing interests.

### Author Contributions

- Ashokkumar Thirunavukkarasu conceived and designed the experiments, performed the experiments, analyzed the data, prepared figures and/or tables, authored or reviewed drafts of the article, and approved the final draft.
- Ahmad Homoud Al-Hazmi conceived and designed the experiments, performed the experiments, prepared figures and/or tables, authored or reviewed drafts of the article, and approved the final draft.
- Umar Farooq Dar conceived and designed the experiments, performed the experiments, prepared figures and/or tables, authored or reviewed drafts of the article, and approved the final draft.
- Ahmed Mohammed Alruwaili analyzed the data, authored or reviewed drafts of the article, and approved the final draft.
- Saleh Dhifallah Alsharari performed the experiments, analyzed the data, authored or reviewed drafts of the article, and approved the final draft.
- Fahad Adel Alazmi conceived and designed the experiments, performed the experiments, analyzed the data, authored or reviewed drafts of the article, and approved the final draft.
- Saif Farhan Alruwaili analyzed the data, authored or reviewed drafts of the article, and approved the final draft.
- Abdullah Mohammed Alarjan conceived and designed the experiments, performed the experiments, analyzed the data, authored or reviewed drafts of the article, and approved the final draft.

### Human Ethics

The following information was supplied relating to ethical approvals (i.e., approving body and any reference numbers):

The Research Ethics Committee Qurayyat Health Affairs approved this study (Approval number-116).

### Data Availability

The raw data is available as a Supplemental File.

## Supplemental Information

Supplemental information for this article can be found online at http://dx.doi.org/10.7717/peerj.13773#supplemental-information.

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
