# Peer review of "Knowledge, attitude and practice towards bio-medical waste management among healthcare workers: a northern Saudi study"

_PeerJ, doi:10.7717/peerj.13773_

## Round 0.1 · original submission · Major Revisions

Authors should address the comments made by the reviewers. Particular attention should be given to the methodology and the justification of the methodological design of the study. In particular, the authors should explain and justify the use of new categories not included in the original database ( grouping of categories) and how this may affect the analysis of the results they have performed.

·

Basic reporting

The researchers have used clear English throughout the manuscript; however, I suggest revising carefully for language and syntax errors. Below are a few suggested edits that could contribute toward more clarity and consistency.

Abstract:
Consider rephrasing the methodology section starting with referring to the current study (tool) then describing sampled population:
e.g: The present study assessed KAP through a self-administered and validated questionnaire. Using a multistage probability sampling method, 384 HCWs from different healthcare facilities participated in this survey. We cross-tabulated the knowledge, attitude, and practice category to find the associated factors with the sociodemographic variables. Spearman's correlation test was performed to find the strength and direction of correlation (rho) between KAP scores.

Minor edits throughout the manuscript:
Inconsistent spacing across the document:
e.g: Line 84: extra space “and / or”
line 107: add "the" before "failure"
Inconsistent hyphenation---- sociodemographic sometimes used and sometimes sociodemographics.

Experimental design

Health care workers are the first in line in managing the BMW. Therefore, assessing BMW's knowledge, attitude, and practices (KAP) is vital to ensure that BMW is properly managed and take the necessary actions if the inadequacies are identified, so the risk to healthcare worker and the general population is at a minimum. Accordingly, the research question of this study is well defined and relevant.
This paper contributes to that knowledge base; however, more detail is needed when describing the methodology of the research and more consideration of how the work links to the literature.

Below are some comments and suggestions for improvement :

Survey questionnaire:
line 170: The researchers need to verify if the tool adapted from the previously published study is a validated tool, as the referenced study methodology does not indicate that their tool is validated. It was only tested for internal consistency (reliability).
The researchers need to provide more details about the final tool they used and the modifications they applied to the original instrument and explain why they were made. Explain the type of questions used to assess knowledge. It would be helpful to provide the survey as supplemental material.

Line 180: Please, provide literature that supports the information referenced. Please, describe the "other "category of healthcare workers.

Statistical analysis section:
Please, make sure you name the statistical test when testing the association of the subscales' categories and sociodemographic characteristics.

Validity of the findings

I want to thank the researchers for their valuable research and for making the data available, well-labeled, and organized. The statistical analysis looks sound; however, I would recommend further analysis that could put more insight into your study finding and how it can be used later in designing BMW educational programs (please, see additional notes). I would recommend sharing the data in an Excel file for easier accessibility.
Below are some comments on the results and the discussion section:

Results section:
Lines 219- 222 that explain table 4 could use more details on the significant findings.
Tables:
Table 1: add a footnote that describes other healthcare worker category
Table 2: Researchers should provide a rationale for grouping low and medium score categories in the methods section. This should be noted in the methods section.
A footnote that describes the statistical test used and used statistical significance level (alpha) is missing.
Tables 3, 4, 5 I would recommend including subscale name (knowledge, attitude, or practice) in the table heading for more clarity.
Please. Ensure all tables include the (n) total number of participants; this can be done in the tile or within the table.

Discussion:
The researchers need to revise and rewrite the discussion. If possible, incorporate the studies' findings, including numbers/statistics, while including your study's findings.

Additional comments

I would recommend a graph that plots KAP subscales’ categories to visualize differences better.
If at all possible, run a regression model to find significant predictors of the different subscales.
I would suggest testing the KAP correlations by work setting and work experiences and seeing if the same patterns will be observed.
I would recommend that the researchers provide some background information on the policies and regulations, for training healthcare workers about BMW management and what educational programs are in place in the studied healthcare settings, its availability, and if mandatory or not, and so. That would be helpful to interpret the results, especially if a difference is noted with nationality type and duration of work (experience).

Also, it would be useful to provide information on how the study finding was communicated with concerned authorities and the organization involved and subsequent actions that were taken or considered to address the issue if any.


According to the study finding, marital status is significantly associated with the behavior score; could the researchers explain this finding? Is there literature available that supports testing this aspect? Support this finding?

Otherwise, well written.
Well done on your work in this important area. Your research covers some really important aspects of this topic and I hope that it will provoke further research in this area – from your team and from others.

Reviewer 2 ·

Basic reporting

-Clear and unambiguous, professional English used throughout: please for article English drafting
-Literature references, sufficient field background/context provided: yes
-Professional article structure, figures, tables. Raw data shared: Methods section needs more depth
-Self-contained with relevant results to hypotheses: yes

Experimental design

-Original primary research within Aims and Scope of the journal: yes
-Research question well defined, relevant & meaningful. It is stated how research fills an identified knowledge gap: yes
-Rigorous investigation performed to a high technical & ethical standard: needs more explanation.
-Methods described with sufficient detail & information to replicate: inadequate

Validity of the findings

-Impact and novelty not assessed. Meaningful replication encouraged where rationale & benefit to literature is clearly stated: yes
-All underlying data have been provided; they are robust, statistically sound, & controlled: some results need more discussion
-Conclusions are well stated, linked to original research question & limited to supporting results: yes

Additional comments

Title: please remove "multicenter study"

Annotated reviews are not available for download in order to protect the identity of reviewers who chose to remain anonymous.

---

## Round 0.2 · Minor Revisions

The authors have put a great deal of effort into revising their manuscript. Some minor aspects of the methodology still need to be clarified. Please clarify within the regression analysis which variables have significant results with respect to the control, and in this respect pay particular attention to show which categories are involved when the variable has more than one category.

·

Basic reporting

Dear Editor,

Generally, the authors have responded appropriately to my comments. They addressed all my concerns and suggestions.

Again, this work is valuable and contributes to the literature.

I have just one comment on how the regression analysis results are presented. The authors did not clearly state which categories/groups of the predictor variables (e.g., age ) have significant differences from the reference group (higher odds of the outcome), especially for variables with three categories. I would recommend using more clear language in describing the results of the regression table, clearly stating what groups have higher odds than which group.

Experimental design

No comment

Validity of the findings

No comment

Additional comments

I want to thank the authors for responding to all my comments, considering my suggestions, and providing a clear explanation for my questions. I enjoyed reading this work, and I think it has added value to the literature and research community.

---

## Round 0.3 · Minor Revisions

When reporting the values associated with the correlation coefficients, the authors have included in the text (e.g., lines 293-295 and in the abstract) a symbol that may suggest a negative value of Spearman's rho coefficient. This should be corrected so as not to mislead since according to the text and Table 6 the correlation is positive.

---

## Round 0.4 · accepted · Accept

The authors have substantially improved their original manuscript and in its current version it is recommended for publication.